# Keep Your Head Up—Correlation between Visual Exploration Frequency, Passing Percentage and Turnover Rate in Elite Football Midfielders

**DOI:** 10.3390/sports7060139

**Published:** 2019-06-06

**Authors:** Ashwin Phatak, Markus Gruber

**Affiliations:** 1Department of Sport Science, University of Konstanz, 78464 Konstanz, Germany; 2Human Performance Research Centre, Department of Sport Science, University of Konstanz, 78464 Konstanz, Germany; m.gruber@uni-konstanz.de

**Keywords:** visual search strategy, visual exploratory frequency (VEF), scans, transition scans, passing percentage, turnovers

## Abstract

Statistical analysis of real in-game situations plays an increasing role in talent identification and player recruitment across team sports. Recently, visual exploration frequency (VEF) in football has been discussed as being one of the important performance-determining parameters. However, until now, VEF has been studied almost exclusively in laboratory settings. Moreover, the VEF of individuals has not been correlated with performance parameters in a statistically significant number of top-level players. Thus, the objective of the present study was to examine the relationship between VEF and individual performance parameters in elite football midfielders. Thirty-five midfielders participating in the Euro 2016 championship were analyzed using game video. Their VEF was categorized into scans, transition scans, and total scans. Linear regression analysis was used to correlate the three different VEF parameters with the passing percentage and the turnover rate for individual players. The linear regression showed significant positive correlations between scan rate (*p* = 0.033, $R^{2}$ = 3.0%) and total scan rate (*p* = 0.015, $R^{2}$ = 4.0%) and passing percentage but not between transition scan rate and passing percentage (*p* = 0.074). There was a significant negative correlation between transition scan rate and turnover rate (*p* = 0.023, $R^{2}$ = 3.5%) but not between total scan rate (*p* = 0.857) or scan rate (*p* = 0.817) and turnover rate. In conclusion, the present study shows that players with a higher VEF may complete more passes and cause fewer turnovers. VEF explains up to 4% of variance in pass completion and turnover rate and thus should be considered as one of the factors that can help to evaluate players and identify talents as well as to tailor training interventions to the needs of midfielders up to the highest level of professional football.

## 1. Introduction

There has been a rise in the use of player performance statistics for the recruitment of players in modern football across professional leagues in Europe and North America [1]. Individual performance statistics such as goals per game, key passes, pass completion rate, tackles won, turnovers, etc. have been used to measure, scout, and define talent. Technical, cognitive, and psychological skills and their contribution to positioning specific skill requirements have been extensively researched as indicators of performance. Athletic ability, coordination, and decision making are considered key performance aspects in both team and individual sports. Going one step further, position-specific skills and specializations have been studied using various on-field and analytical studies.

Skill requirements and degree of mastery vary from position to position. When it comes to football, certain positions require specific traits [1]. Center midfielders (attacking and defensive) are required to pick up and process information from multiple sources since they spend most of the time at the central parts of the field and are required to have a 360-degree visual input for optimal performance [2,3,4]. Midfielders are required to make quicker decisions in limited space and time on and off the ball compared to other positions.

Team sport involves complex dynamic situations, and variables in the form of players, conditions, tactics, etc. The aspect of the correct and high speed of decision making is present and is considered an indicator of performance across sports despite fundamental differences in both players and sports. There is accumulating evidence that skilled perception is a key aspect of decision making in sport performance [5]. Studies have shown that professional athletes have enhanced abilities to rapidly learn complex and neutral dynamic visual input. Football involves 22 players in a constantly changing dynamic environment [6,7]. Expert football players show an enhanced ability to recognize, process, interpret, and act upon visual input compared to their novice counterparts. The anticipation of play progression seems to be the key. Expert players show a significantly better understanding of “when” and “where” to look in an ever-changing in-game environment [8]. Even if less skilled players know “when” and “where” to look, they might be forced to keep their eye on the ball due to limited technical skills [9]. Higher skilled players showed a greater level of automation compared to their less skilled counterparts in soccer-specific technical abilities, enabling them to direct attention and visual resources to the secondary task of visual exploration [2,9,10]. 

“Visual search strategy” is the way in which performers continually move their eyes to focus on important features on a screen enabling them to filter relevant information and act upon it [11]. A set of previous studies have examined visual search strategy mainly in laboratory settings using screens of different sizes and eye-tracking equipment. It has been shown that the number of fixations on the screen is higher in experienced soccer players, while the fixation duration is shorter compared to their novice counterparts [5]. Skills important to midfielders such as playing a forward pass, passing to an area opposite to where the ball was received from, turning with the ball, and playing a one-touch pass all showed associations with visual exploration [12]. 

Studies conducted on the field or using video analysis are restricted to specific situations in which the observed midfielders are receiving the ball from the defender [13]. Only restricted research in very specific game situations has been performed on the visual search strategy in a real game situation with a statistically significant number of players [13]. Previous studies were done in a situation only when the concerned players are in a certain situation on the field viz. a midfielder receiving the ball form the defender.

Visual exploratory frequency (VEF), an important variable in visual search strategy is the frequency of body and/or head rotational movement prior to receiving the ball, engaged in perceiving information away from the ball [14]. The current study examined a normally distributed set of players over a period of a month and examined their VEF when their team was in possession of the ball. Furthermore, it examined the correlation of scan rates with the individual performance statistics of the respective players.

In the present study, we correlated VEF with individual performance statistics of midfielders from teams that participated in the knockout stages of the Euro 2016 championship. We hypothesized that there would be a positive correlation between VEF and average pass completion rate. Along with a negative correlation between VEF and turnover rate. 

## 2. Materials and Methods

### 2.1. Study Design

We obtained footage from all the 51 games of Euro 2016 football championships via Wyscout (https://wyscout.com/, April 2018) and player performance statistics via whoscored.com (https://www.whoscored.com/Regions/247/Tournaments/124/Seasons/4246/International-European-Championship), which accesses its data from Opta sports (https://www.optasports.com). Thirty-five male center midfielders with a mean age of 29 ± 3 years were then selected based on three criteria. We selected center midfielders (see Appendix A) only from teams that entered the round of the last sixteen, who played more than 4 games and at least 250 minutes in the whole tournament. The participants were from various countries of Europe and had diverse cultural and racial backgrounds. After selection of the players, the footage was analyzed using a PC, Dell, Inspiron 13 core i-7, Konstanz, Germany) and VLC media player version 3.0.4. All selected players were observed across all games in the group and knockout stages over the whole tournament. The observations were divided into 6 (+2 in case of overtime) parts in each game viz. 0–15, 15–30, 30–45, 45–60, 60–75, and 75–90 min, according to periods of game time. In case of substitution, players were only observed during their on-field playing time, and in cases of overtime, the two overtime periods were also analyzed. Players were observed up to a maximum of 90 ± 5 seconds in the respective periods only when their team was in ball possession and the players were visible on the footage.

We defined a scan based on the definition of Jordet, as a body and/or head movement where the observed players look away from the ball with an active rotational neck movement in dynamic gameplay situations except while receiving the ball as shown in Figure 1a [14]. A transition scan was defined as a scan that was performed while receiving the ball, after the ball had already left the previous passer’s foot and was in transition to the observed player as shown in Figure 1b. The third parameter was total scans, which is the sum of scans and transition scans. Scans, transition scans, and total scans served as independent variables for the regression analysis.

By differentiating the total scans in the above categories, we aimed at isolating the scans that were conducted under the additional requirement of performing a technical skill (controlling the ball). It can be assumed that players with a better technical skill may be more comfortable scanning even when the ball is in transition (in the case of ‘transition scan’), as it has been shown that additional cognitive tasks caused a greater reduction in technical performance of novice players compared to experts [10]. 

The obtained stats consisted of average passing percentage in the tournament, which was defined as “percentage of attempted passes that successfully found a teammate”, and turnovers per minute, which stood for “total number of loss of possession due to a mistake/poor control per minute of time played in the tournament”. The passing percentage and the turnover rate as performance parameters served as dependent variables for the regression analysis (Definitions taken form https://www.whoscored.com/Glossary).

The number of “scans” and “transition scans” performed by the player were counted in the abovementioned observation criteria and were normalized by the total number of seconds the player was observed (Scans/second for every game). The VEF (both scan and transition scan) rate was calculated. The values of “scans” and “transition scans” were added and normalized in the above fashion to get a third variable named “total scans”. This procedure ensured a normal distribution of points gathered from observations for a time period of over a month which could account for varying opponents, stage of the game, and any other external conditions while maintaining the realistic in-game observation scenario.

### 2.2. Statistics

We calculated linear regressions between the independent variables (scans, transition scans, and total scans) and the dependent variables (passing percentage and turnover rate). The significance level was set at *p* < 0.05. We used R-squared to calculate the proportion of variation in the dependent variables that is explained by our regression model. We used R-Studios version 3.5.2 to perform the linear regression analysis and calculate *p*- and *R*^2^ values.

### 2.3. Interrater Reliability

Five games from five different midfielders were chosen randomly and analyzed by two different experimenters in order to check the interrater consistency of “transition scans” and “scan” count (see Appendix B). 

The ratio scale level data in the current study were analyzed using the “Krippendorff’s Alpha” as an interrater reliability test. Variables at different levels of measurement and consideration of agreements were well accounted in the 0.00 to 1.00 reliability scale. Here, the reliability increases from 0.00 (no reliability) to 1.00 (perfect reliability) [15]). The reliability analysis was done using IBM PC and SPSS Statistics software with “k-alpha” as the external macro. The Krippendorff’s alpha reliability test showed alpha values of 0.73 for “scans” and 0.86 for “transition scans”. 

## 3. Results

### 3.1. Average Passing Percentage vs VEF

Results below in Figure 2 and Table 1 show a positive correlation between all three types of scan and the average passing percentage of the selected players. Out of the three, scans and total scans show a significant positive relationship (*p* = 0.033, $R^{2}$ = 3.0% and *p* = 0.015, $R^{2}$ = 4.0% respectively) while the transition scans show no significant relation (*p* = 0.074).

### 3.2. VEF vs. Turnovers Per Minute

Results below in Figure 3 and Table 2 show a significant negative correlation between turnovers per minute and “transition scans” (*p* = 0.023, $R^{2}$ = 3.51%). The scans and total scans show no significant correlations.

## 4. Discussion

The first result of the present study was that the scanning rate of a player was positively correlated with his pass completion rate. Although a considerable amount of exceptions exists, a higher pass completion showed tendencies of higher levels of success in the 2014 FIFA world cup [16]. It has been shown in multiple studies that a high pass completion rate in individuals and in teams indicates higher individual and team performance [17,18]. Passing statistics, especially completion rate, may be a key indicator of individual player performance according to the results in the current study. VEF might play a role in reaching high pass completion rates. The present study shows a significant positive correlation between the VEF (total scans and scans) and pass completion rates of midfielders and may be an indicator that average VEF may contribute to individual passing performance in a real game situation, although the $R^{2}$ values shown in correlations were very low (3%–4%). Such differences may still play an important role in determining the ability of an elite player. As in the case of elite youth players, these small differences may result in larger differences over time with deliberate training [19].

The second result of the present study was that the scan rate while receiving the ball was negatively correlated with his turnover rate. Possession in soccer seems to be one of the key factors in deciding match outcomes [18]. The team with players who cause fewer turnovers has a higher possession of the ball. A recent study showed a significant negative correlation between transition scan rate and number of turnovers per minute. However, scan rate per second was not correlated with turnover rate [9]. These results are in accordance with the results of the present study and suggest that the transition scan performed by the player is more important to keep the ball safe (i.e., not causing turnovers) compared to the other two VEF types. The visual search of the player in concern, right before receiving the ball, seems to be crucial. It may provide further evidence confirming the prediction that a player of higher technical ability may need to allocate less visual attention on the ball and allocate more to visual exploratory searches [9]. The low $R^{2}$ values indicated high variance, which may suggest that although VEF is a piece of puzzle in modeling passing and ball retention performance, it is only a very small piece of the entire model. The current study provides further evidence on the positive relationship of VEF on various performance indices as stated in previous studies [4,11,14,20].

### Limitations

One possible source of error may be the lack of split screen footage and the spatial resolution of the videos. Thus, it was sometimes difficult to clearly define a scan, especially when the player was on the far side of the field with respect to the camera. This might also explain why interrater reliability failed to be excellent despite a seemingly easy observation task. A more reliable and objective approach would be to measure the head movement using inertial measurements units (IMU) [12]. However, in the present study, we aimed at analyzing the top-level players in real game situations, which makes it impossible to use IMU technology. Another approach using video analysis would be to use split screen footage and use machine learning algorithms to increase quality, reliability and control for interrater subjectivity. Furthermore, eye movement as a factor for visual exploration was ignored, and thus, it can be that in some players, we underestimated VEF. Finally, it must be acknowledged that even though correlations turned out to be significant, *R*^2^ values were low, questioning their practical relevance. However, as the present study was performed during a competition including the world’s top-class midfielders, even small effects can be key.

## 5. Conclusions

The study results confirmed a positive correlation between VEF and average pass completion rates, as well as a negative correlation between VEF and turnovers per minute for transition scans. Scans and transition scans can thus be used as indicators of a player to complete a high percentage of passes. A higher transition scan rate specifically predicts the ability of a player to not lose the ball, i.e., cause fewer turnovers. It seems to be evident from the results that transition scans may be a more specific parameter for measuring ball retention performance compared to scans when it comes to in-game situations. 

The current findings may provide perspectives for talent identification and improved coaching methods by focusing on increasing VEF in line with previous research. Focus on improving transition scan rate may provide an improvement in passing and ball retention capabilities of midfield players. The method used to determine the VEF in the current study can be used to study other performance indices as it assures, for the most part, statistically significant normal distribution of points in random scenarios of the game. The method can be further used in future studies along with machine learning algorithms, high-quality footage, and big data to provide us with models that allow a more accurate prediction of passing performance.

## Figures and Tables

**Figure 1 sports-07-00139-f001:**
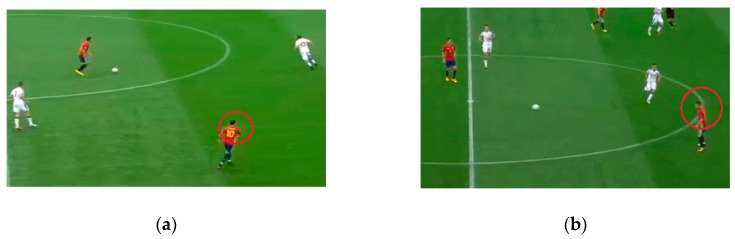
(**a**) Player (red circle) performs a scan by turning his head and looking away from the player in ball possession. (**b**) Player (red circle) performs a transition scan by turning his head and looking away from the player and the ball while the ball is in transition.

**Figure 2 sports-07-00139-f002:**
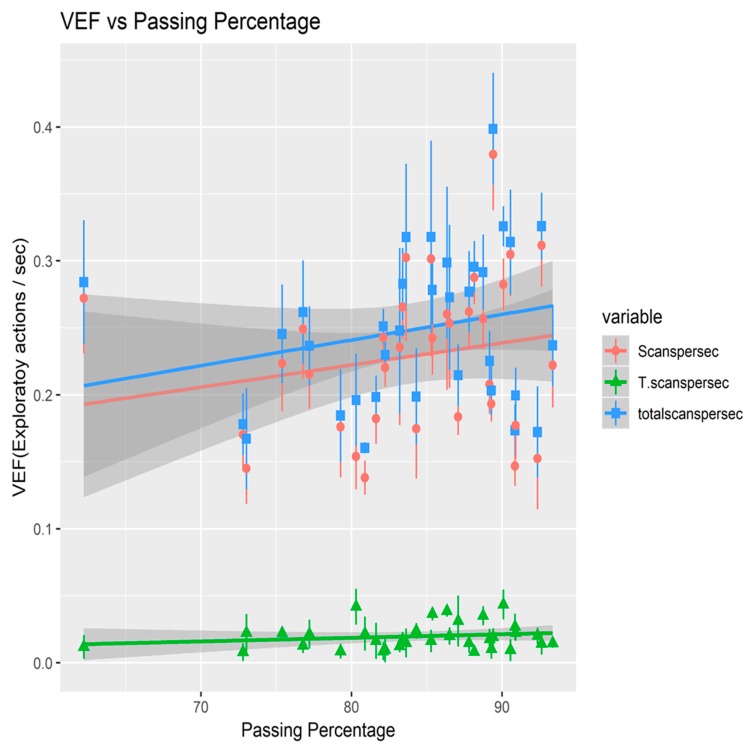
Correlation between the 3 types of VEF vs. the passing percentage.

**Figure 3 sports-07-00139-f003:**
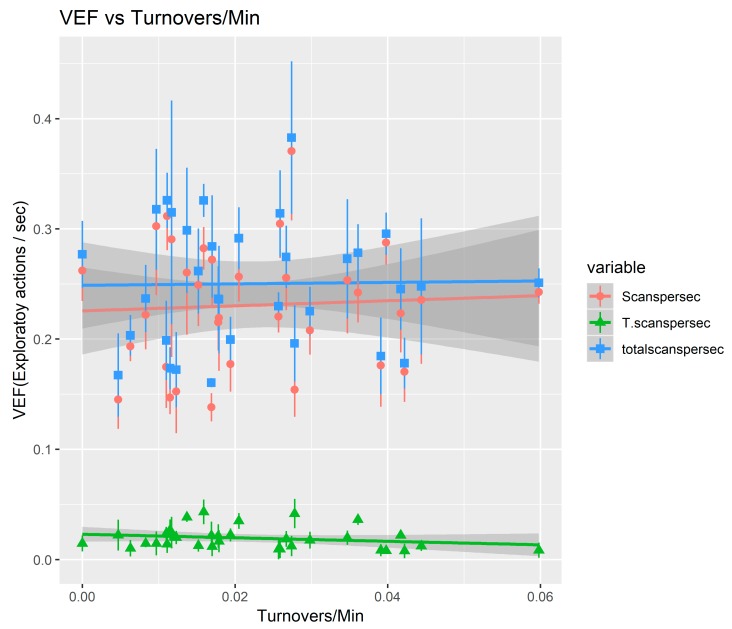
Correlation between 3 types of VEF vs. average turnover rate of players.

**Table 1 sports-07-00139-t001:** Results of the linear regression for VEF vs. passing percentage.

VEF	Estimate	Std. Error	t Value	R Squared (%)	P-Value	F-Value
Scans	16.728	7.836	2.135	3.0%	0.033	4.556
Transition scans	78.694	44.005	1.788	2.1%	0.074	3.168
Total scans	19.181	7.837	2.448	4.0%	0.015	6.049

**Table 2 sports-07-00139-t002:** Results of the linear regression for VEF vs. average turnover rate of players.

VEF	Estimate	Std. Error	t Value	R Squared (%)	P-Value	F-Value
Scans	0.0037	0.0162	0.2314	0.04%	0.8171	0.0529
Transition scans	−0.2066	0.0908	−2.2758	3.51%	0.0233	5.3035
Total scans	−0.0029	0.0162	−00.1802	0.02%	0.8571	0.0321

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
