# Peer review of "Keep Your Head Up—Correlation between Visual Exploration Frequency, Passing Percentage and Turnover Rate in Elite Football Midfielders"

_sports, 2019, doi:10.3390/sports7060139_

Round 1

Reviewer 1 Report

General comments

Please see the following paper on visual exploratory actions in football: McGuckian TB, Cole MH, Jordet G, Chalkley D and Pepping G-J (2018) Don’t Turn Blind! The Relationship Between Exploration Before Ball Possession and On-Ball Performance in Association Football. Front. Psychol. 9:2520. doi: 10.3389/fpsyg.2018.02520

The current paper could benefit greatly from the abovementioned study, in terms of providing the most recent applications of visual exploration in football. Furthermore, whilst the current study is specific to midfield positions, the MuGuckian et al context will improve the paper.

The obvious limitation of the current study is that the scanning is subjective, however, obtaining data using wearable instruments from the best players in the world, in the worlds second biggest tournament is impossible. I think you need to state that there are more sophisticated measurement tools, but there is still a need for the current analysis, as it describes the actions of the elite players.  

Please change the in-text citations to author-year in the following instance ‘According to [1] …….’ It reads better as ‘According to Smith (2000) ………’

Some grammar needs revising

Introduction

The introduction needs to include the McGuckian study to put into context the current work to date.

VEF needs to be defined at first use of the Introduction. Line 74.

Method

Line. 87-88. Good ‘video’ footage

It is not good enough to say ‘sport science journals use the 95% CI’. Please explain why it is used in the context of the current study.

Design

The research design section is unclear and needs additional information. The current design is more like the procedure. Please revise this section to include, the theory used, the rationale to why it was used, and the describe how these elements are implemented in the study. The dependent and independent variables should be clear.

Assessment and Measures.

If the scanning figures are not published in colour you will need to reword (red number 10 etc)

Please add appropriate figure legends, that describe the figures.

Results

Please remove the wording ‘trend towards significance’ If it is not at the level you specified it is not significant.

Figure 2 and 3 resolution is not great, and both need figure legends.

Discussion

Again, the McGuckian study needs to be discussed in the context of the current results. What is lacking in the Discussion is how the current study fits with other studies of visual exploration in ‘real-life’ football matches. As it is, we have no idea how the current results compare to others.

The study needs some practical implications for coaches.

Author Response

General comments

Please see the following paper on visual exploratory actions in football: McGuckian TB, Cole MH, Jordet G, Chalkley D and Pepping G-J (2018) Don’t Turn Blind! The Relationship Between Exploration Before Ball Possession and On-Ball Performance in Association Football. Front. Psychol. 9:2520. doi: 10.3389/fpsyg.2018.02520                                               

The current paper could benefit greatly from the abovementioned study, in terms of providing the most recent applications of visual exploration in football. Furthermore, whilst the current study is specific to midfield positions, the MuGuckian et al context will improve the paper.

Answer: Thanks for the recommendation. Please notice that the study design and data collection of the present study were done prior to the publishing of the above study. Nevertheless, we do agree that indeed the study discusses key elements and provide data that helps to motivate and discuss the results of the present study. Thus, we included and now cite the study of McGuckian et al. in the introduction & discussion and specifically in the limitations section.

The obvious limitation of the current study is that the scanning is subjective, however, obtaining data using wearable instruments from the best players in the world, in the world’s second-biggest tournament is impossible. I think you need to state that there are more sophisticated measurement tools, but there is still a need for the current analysis, as it describes the actions of the elite players.  

Answer: Thank you very much for this note and the arguments that justify our methodological approach in the present study. We updated the limitations section accordingly.

Please change the in-text citations to author-year in the following instance ‘According to [1] …….’ It reads better as ‘According to Smith (2000) ………’

Answer: Noted, The in-text citations & citation style have been revised according to the journal style.

Some grammar needs revising

Answer: We revised the manuscript with special emphasis on grammar.

Introduction

The introduction needs to include the McGuckian study to put into context the current work to date.

Answer: We agree and included the above-mentioned study.

VEF needs to be defined at first use of the Introduction. Line 74.

Answer: Thanks, we now have included the definition of VEF.

Method

Line. 87-88. Good ‘video’ footage

Answer: Modified as suggested

It is not good enough to say ‘sport science journals use the 95% CI’. Please explain why it is used in the context of the current study.

Answer: Methods and study design has been restructured according to the requirements

Design

The research design section is unclear and needs additional information. The current design is more like the procedure. Please revise this section to include, the theory used, the rationale to why it was used, and the describe how these elements are implemented in the study. The dependent and independent variables should be clear.

Answer: The design has been reconstructed to better explain the rationale of the study

Assessment and Measures.

If the scanning figures are not published in colour, you will need to reword (red number 10 etc)

 Answer: Thanks, we Modified figure 1 and included a highlighted circle on the player

Please add appropriate figure legends, that describe the figures.

Answer: Legends added

Results

Please remove the wording ‘trend towards significance’ If it is not at the level you specified it is not significant.

Answer: We agree and removed it, as suggested

Figure 2 and 3 resolution is not great, and both need figure legends.

Answer: Graph resolution increased to 500dpi. Unfortunately, the footage snapshots for scans and transition scans are at the best possible quality available to us.

Discussion

Again, the McGuckian study needs to be discussed in the context of the current results. What is lacking in the Discussion is how the current study fits with other studies of visual exploration in ‘real-life’ football matches. As it is, we have no idea how the current results compare to others.[mg1] 

Answer: Has been included in Limitations and discussions. Along with the R squared value have been explained as to how they may be crucial at the elite level.

The study needs some practical implications for coaches.

Answer: Practical suggestions have been provided in the discussion section

Reviewer 2 Report

 The paper is well written and organized. In most cases, it is comprehensive. Because of that and the fact that it deals with the analysis of sports, it is a real pleasure to read it. However, there are some issues, which should be addressed:

·         The language and the text format must be revised, e.g. remove underlined text phrases, remove duplicated abbreviations, etc.

·         Please also improve the images in terms of scale and resolution.

·         Can you give examples images, which show the corresponding player behavior (scanning)?

·         I think in many cases players can also scan their environment just by moving their eyes without turning the head. I doubt that those situations are considered in this work (because you will not see it in the frames.). However, those scans may change the performance values as well!?

Further interesting additions to this and consecutive papers would be:

·         Extract the scans of the players automatically by using methods of computer vision and deep learning?

·         May be an important question would also be whether the players look at the right places at the right time. And: what are those right places?

Author Response

The paper is well written and organized. In most cases, it is comprehensive. Because of that and the fact that it deals with the analysis of sports, it is a real pleasure to read it. However, there are some issues, which should be addressed:

The language and the text format must be revised, e.g. remove underlined text phrases, remove duplicated abbreviations, etc.

Answer: Thank you for your positive feedback on our study and manuscript. According to your suggestion, we edited language with a special focus on grammar and style and hope that it is now appropriate for publishing.

Please also improve the images in terms of scale and resolution.

Answer: Graph resolution increased to 500dpi. Unfortunately, the footage stills are for scans and transition scans are as good as they get

Can you give examples images, which show the corresponding player behaviour (scanning)?

Answer: Fig 1a,1b have been modified to circle the player in concern

 I think in many cases players can also scan their environment just by moving their eyes without turning the head. I doubt that those situations are considered in this work (because you will not see it in the frames.). However, those scans may change the performance values as well!?

Answer: Yes, excellent point. This is definitely a limitation of the study and we have added that on line 230.

Further interesting additions to this and consecutive papers would be:

 Extract the scans of the players automatically by using methods of computer vision and deep learning?

Answer: Again, an excellent point. That was also our initial idea. However, there is a lack of good quality footage and cameras are not always well positioned for kinematic analysis. Moreover, ML algorithms are not yet up to the mark for such small movements. In future research (PhD of Ashwin Phatak) we plan to use ML and position data for analysis.

Maybe an important question would also be whether the players look at the right places at the right time. And: what are those right places?

Answer: 

Good point. We included a study in the introduction (line 58,59) that address this issue

Ref.: M. Williams and K. Davids, “Visual search strategy, selective attention, and expertise in soccer,” Res. Q. Exerc. Sport, vol. 69, no. 2, pp. 111–128, 1998.

Reviewer 3 Report

In this study the relationship of head turning and passes were assess. Public available videos of soccer games were manually tagged. Here the authors also provide an inter-tester reliability study. While in general the study is well conducted and presented, there are some few points that should be addressed.

-        It is unclear to the reader why this study with manual tracking should relate to “Big Data”. Nowadays people develop automated tagging algorisms. Here a rather solid but old school approach was used.

-        Please include the inter-tester reliability in the aim of the study.

-        It is not clear how the 33 participants were calculated. The names in appendix A and in line 152-155 are not equal to 33. Why are some players in the text and others in the appendix? I would recommend making a table only in the appendix and presenting all of them there.

-        Please present the total amount of head turn in the result section.

-        Please remove the statement 213-215. This is probably part of the instructions.

-        207. Please remove .

-        Table 2. Please align the values “estimate”

Author Response

It is unclear to the reader why this study with manual tracking should relate to “Big Data”. Nowadays people develop automated tagging algorisms. Here a rather solid but old school approach was used.

Answer: You are right, we removed the notes on “Big Data” from the present manuscript as it can be misleading.

-        Please include the inter-tester reliability in the aim of the study.

Answer:  Thank you for this suggestion. However, to assess the inter-tester reliability was not a specific scientific aim of the present study. It was more to ensure that our methodological procedure was reliable. To provide a clear aim and to not entangle the reader we didn´t include the inter-tester reliability in the aim of the study. If you still feel that this is necessary, please let us know.  

-        It is not clear how the 33 participants were calculated. The names in Appendix A and in

line 152-155 are not equal to 33. Why are some players in the text and others in the appendix? I would recommend making a table only in the appendix and presenting all of them there.

Answer:  You are right, Thanks for pointing out this mistake. There was a typo which has been corrected. We analyzed 35 players. According to the suggestion of another reviewer, we removed the names of the analyzed players from the manuscript and moved them in the appendix.

-        Please present the total amount of head turn in the result section.

Answer: Thanks for the suggestion but the results of head turn section will be added in the supplementary materials according to the requirements of the journal as the table is very large to include in the paper itself.

-        Please remove the statement 213-215. This is probably part of the instructions.

Answer:  Have been removed as requested

-        207. Please remove.

Answer:  Has been removed as requested

-        Table 2. Please align the values “estimate”

Answer:  We revised the figures and tables and aligned the values.

Reviewer 4 Report

General comments.

The introduction and the methods section of the article were very well written and enjoyable to read, although the numbered references occasionally made some sentences quite difficult to understand. It is refreshing to see that the more open-play situations are being investigated as opposed to the closed-skill, set-plays of the past. However, the results section needed some more explanation and some specific data values were missing (namely actual R or R squared values). The discussion section was rather too brief and did not really fully explore the implications of the findings. Lots of may and might with VEF (the title and hypothesis focus) was only mentioned in the final paragraph. A better and more substantial link to the talent identification and implications for performance could be established in the discussion and conclusion section.

Line 56 – There is a change in referencing format, using an author’s name instead of the numbering system.

Lines 68-76 – In this section the wording and explanation is very difficult to follow. The gist of linking previous studies to this one in order to justify its importance can be deduced but this is not explicit enough in the actual writing. In particular it is unclear which situations have been used in previous work; open-play or set-piece game situations?

Lines 77-80 – In this section it was unclear where this hypothesis came from, perhaps in the literature previously explored this could be made more prominent.

Line 83 – it is usual to report the standard deviation data to enable the reader to make a judgement about the age range of participants.

Lines 109-135 - The assessments and measures section was very clear and easy to understand how the data was obtained.

Lines 152-156 – The actual players’ names should not be reported in this work even from secondary, publicly available data, this is not ethical practice and breaches confidentiality and anonymity of participants. The inter-rater reliability value can and should be reported in this section to enable the reader to establish the validity of the process.

Lines 164-167 – whilst reporting p values for correlations is good practice, reporting the r value is more important to establish the strength of the relationship.

Table 1 data needs some further explanation. The R squared value is usually expressed as a percentage showing how much variation is explained through the measured construct. The R values are almost more important than the p values as it shows the strength of the relationship.

Lines 182-183 – This sentence would have more impact if the percentage of variation (r squared) was also reported, is this a low, medium or high correlation leading to low, medium or high explained variance?

Lines 186-187 – be careful about making this assumption as not all games with high possession (indicating high pass rate) lead to successful outcomes.

Line 210 – this is the first time that the article admits to low R squared values. Therefore, explaining some of the caution with interpretation of the results.

Line 213 - there seems to be some formatting cues at the start of this line.

Author Response

The introduction and the methods section of the article were very well written and enjoyable to read, although the numbered references occasionally made some sentences quite difficult to understand. It is refreshing to see that the more open-play situations are being investigated as opposed to the closed-skill, set-plays of the past. However, the results section needed some more explanation and some specific data values were missing (namely actual R or R squared values).

The discussion section was rather too brief and did not really fully explore the implications of the findings. Lots of may and might with VEF (the title and hypothesis focus) was only mentioned in the final paragraph. A better and more substantial link to the talent identification and implications for performance could be established in the discussion and conclusion section.

Answer: intensive modifications have been made to the discussion and additional study is included along with further clarifications

Line 56 – There is a change in referencing format, using an author’s name instead of the numbering system.

Answer: Has been addressed  according to the requirements of the journal

Lines 68-76 – In this section, the wording and explanation are very difficult to follow. The gist of linking previous studies to this one in order to justify its importance can be deduced but this is not explicit enough in the actual writing. In particular, it is unclear which situations have been used in previous work; open-play or set-piece game situations?

Answer: The paragraph has been revised to make the previous studies better link to the hypothesis

Lines 77-80 – In this section, it was unclear where this hypothesis came from, perhaps in the literature previously explored this could be made more prominent.

Answer: The background is provided in the previous 2 paragraphs from line 70 to line 85. I have added an extra reference to drive the point home.

T. B. Mcguckian, M. H. Cole, G. Jordet, and D. Chalkley, “Don ’ t Turn Blind ! The Relationship Between Exploration Before Ball Possession and On-Ball Performance in Association Football,” vol. 9, no. December, pp. 1–13, 2018

Line 83 – it is usual to report the standard deviation data to enable the reader to make a judgement about the age range of participants.

Answer: Noted and has been added in the participants' section

Lines 109-135 - The assessments and measures section were very clear and easy to understand how the data was obtained.

Lines 152-156 – The actual players’ names should not be reported in this work even from secondary, publicly available data, this is not ethical practice and breaches confidentiality and anonymity of participants.

Answer: They are removed for the text and kept only in the appendix. We will further follow the recommendations of the Editor.

 The inter-rater reliability value can and should be reported in this section to enable the reader to establish the validity of the process.

Answer: Thank you for this suggestion. However, to assess the inter-tester reliability was not a specific scientific aim of the present study. It was more to ensure that our methodological procedure was reliable. To provide a clear aim and to not entangle the reader we didn´t include the inter-tester reliability in the aim of the study. If you still feel that this is necessary, please let us know.  

Lines 164-167 – whilst reporting p values for correlations is good practice, reporting the r value is more important to establish the strength of the relationship.

Answer: Has been added to results and the abstract

Table 1 data needs some further explanation. The R squared value is usually expressed as a percentage showing how much variation is explained through the measured construct.

Answer: It has been converted to percentage values

The R values are almost more important than the p values as it shows the strength of the relationship.

Answer: R squared values mentioned in discussion, conclusion, results and also added to the abstract text.

Lines 182-183 – This sentence would have more impact if the percentage of variation (r squared) was also reported, is this a low, medium or high correlation leading to low, medium or high explained variance?

Answer: Noted and elaborated in discussion section line 227

Lines 186-187 – be careful about making this assumption as not all games with high possession (indicating high pass rate) lead to successful outcomes.

Answer: Modified wording for better communication line 211

Line 210 – this is the first time that the article admits to low R squared values. Therefore, explaining some of the cautions with the interpretation of the results.

Answer: I have modified it and mentioned it in the discussion, also addressed it more in limitations.

Line 213 - there seem to be some formatting cues at the start of this line.

Answer: Has been removed

Round 2

Reviewer 1 Report

Thank you for making the suggested changes, I believe these have strengthened the manuscript. 

Author Response

Thank you for making the suggested changes, I believe these have strengthened the manuscript. 

Answer: Language and grammar have been updated as suggested

Reviewer 3 Report

Thanks for the revision. Well done. Please use the same number of digits for the value and the standard deviation. eq. 29pm3.

Author Response

Thanks for the revision. Well done. Please use the same number of digits for the value and the standard deviation. eq. 29pm3.

Answer:  The suggested change has been included in the statistics section